# Long-Time Non-Debye Kinetics of Molecular Desorption from Substrates with Frozen Disorder

**DOI:** 10.3390/molecules25163662

**Published:** 2020-08-11

**Authors:** Victor N. Bondarev, Volodymyr V. Kutarov, Eva Schieferstein, Vladimir V. Zavalniuk

**Affiliations:** 1Research Institute of Physics, I.I. Mechnikov National University, 27 Pasteur St., 65082 Odessa, Ukraine; bondvic@onu.edu.ua (V.N.B.); v.kutarov@onu.edu.ua (V.V.K.); 2Fraunhofer UMSICHT, 3 Osterfelder Str., D-46047 Oberhausen, Germany; 3Department of Fundamental Sciences, Odessa Military Academy, 10 Fontanska Road, 65009 Odessa, Ukraine; vzavalnyuk@onu.edu.ua

**Keywords:** desorption, disordered adsorbents, non-Debye kinetics, fluctuation theory

## Abstract

The experiments on the kinetics of molecular desorption from structurally disordered adsorbents clearly demonstrate its non-Debye behavior at “long” times. In due time, when analyzing the desorption of hydrogen molecules from crystalline adsorbents, attempts were made to associate this behavior with the manifestation of second-order effects, when the rate of desorption is limited by the rate of surface diffusion of hydrogen atoms with their subsequent association into molecules. However, the estimates made in the present work show that the dominance of second-order effects should be expected in the region of times significantly exceeding those where the kinetics of H_2_ desorption have long acquired a non-Debye character. To explain the observed regularities, an approach has been developed according to which frozen fluctuations in the activation energy of desorption play a crucial role in the non-Debye kinetics of the process. The obtained closed expression for the desorption rate has a transparent physical meaning and allows us to give a quantitative interpretation of a number of experiments on the desorption kinetics of molecules not only from crystalline (containing frozen defects) but also from amorphous adsorbents. The ways of further development of the proposed theory and its experimental verification are outlined.

## 1. Introduction

Molecular adsorption-desorption processes belong to one of the most important sections of physicochemical phenomena occurring at interphase boundaries [1,2]. Recent interest in the study of these processes is stimulated by the introduction of new adsorbents with optimal properties in relation to the accumulation and release of different molecules (in particular, water). Currently, there is fairly extensive literature on the kinetic properties of microporous materials like zeolites and metal-organic framework structures, which demonstrate good characteristics of the adsorption-desorption processes (see, for example, review articles [3,4,5]). 

However, a consistent analysis of the obtained experimental data on the molecular desorption from real adsorbents (including those mentioned in the above references) has so far been difficult. Indeed, in most cases, the kinetics of the process do not obey the expected laws, for example, under the assumption of desorption as a “chemical” reaction of the first order. Attempts to describe the kinetics of desorption of *n*-alkane molecules from zeolites (e.g., NaX, [2]) by a simple Debye exponential on time have not yielded results. The reason for this circumstance, in principle, could be due to the fact that the zeolites used in the experiments [2] have a branched internal structure, so the kinetic characteristics of the adsorbed molecules leaving the “external” surface of the zeolite and the pore area in the volume would differ. Thus, to describe experiments like [2], it would be necessary to use at least two Debye exponentials. However, this does not mean that an exhaustive interpretation of the experimental dependences [2] can be given in this way: The long-term desorption kinetics inevitably deviates from the predictions of such a simplified scheme (see below).

Meanwhile, to date, the results of experiments on the molecular desorption from the surface of crystalline adsorbents are known. For example, in papers [6,7,8,9,10,11,12] one can find information concerning the desorption kinetics of hydrogen from the crystalline silicon surfaces. In this case, to describe the dependence of the hydrogen desorption rate on time *t*, the so-called Polanyi-Wigner kinetic equation was used (see, e.g., [9]):
(1)$$\frac{d\theta }{dt}=-k_{m}\theta^{m}$$
where *θ* is the relative fraction of the coverage of the silicon surface with hydrogen, *m* is the order of desorption as a “chemical” reaction and $k_{m}$ is the corresponding rate constant. Note that the adsorbent coverage studied in the experiments [9], did not exceed one adsorbate monolayer, i.e., *θ* ≤ 1, which we will assume in what follows. Obviously, the value of *θ* at an arbitrary time moment is related to the average area *a*^2^ per one hydrogen atom at the same time moment, by the relation $\theta =a_{0}^{2}/a^{2}$ where $a_{0}^{2}$ is the area per one hydrogen atom at the maximum monolayer filling (in the case of Si(100)2×1, this area is estimated by the value $a_{0}^{2}$ = 1.47 × 10^−15^ cm^2^ [11]).

In due time, a discussion even started in the literature [6,7] about the value of *m*, which describes the process of desorption of hydrogen molecules from the surface of a silicon crystal. For example, in [6], the hydrogen desorption rate was described by the second-order reaction, in fact, based on the assumption that the surface diffusion of hydrogen atoms with the subsequent formation of H_2_ molecules is the rate-limiting step of the desorption process. On the contrary, in [7], the experimental dependences of hydrogen desorption on time were approximated, based on the model of the first-order “chemical” reaction with respect to the surface concentration of H_2_ molecules. Besides, the first-order reaction equation was used in [8] to describe the rate of hydrogen desorption from the surface of a germanium crystal. Further, in [9], experimental data on hydrogen desorption from the surface of Si(111)7×7 adsorbent (“lightly doped with phosphorus”) were presented. It was found [9] that the long-term (>1000 s at temperature 720 K) kinetics of hydrogen desorption cannot be described in terms of “chemical” reactions of the first or second order. On the other hand, the use of an effective desorption exponent *m* ≈ 1.56 made it possible to carry out a formal fitting of the experiments [9], although the physical interpretation of this result remained unclear. Later in [10,11], a model expression was proposed for the desorption rate, which allows a formal transition from the first-order reaction at “large” *θ* to the second-order reaction at “small” *θ*. Similar results on hydrogen desorption from Si(100) surfaces were obtained in [12] using a more complex scheme of kinetic equations. It is significant, however, that such a way, in principle, could not lead to a fractional exponent (*m* ≈ 1.6 [10,11,12]) for the long-term desorption kinetics.

In the above examples, we, in fact, dealt with the so-called chemisorption characterized by the activation energy of hydrogen desorption at the level of 50–60 kcal/mol (2.2–2.6 eV) [7]. The aforementioned change in the desorption rate from first to second order with decreasing *θ*, in fact, was completely associated with the features of the formation mechanism of H_2_ molecules on the adsorbent surface [10,11,12]. In this case, however, it remained unclear to what extent the effects of structural disorder of the adsorbent surface, which is subject to fluctuations in composition, can affect the resulting rate of molecular desorption. As shown in experiments [13], the crystal surface Si(001) has a high density of vacancy-type defects. Recall that, in addition to intrinsic defects, in almost all cases, crystalline adsorbents contained a certain fraction of impurities, for example, phosphorus in Si [9] or gallium in Ge [8]). In pure form, an answer to a similar question could be obtained if one and the same structural units, primarily atoms of one and the same substance, participated in the processes of adsorption and desorption. In other words, if the step of formation of molecules from adsorbed atoms would not be required for the implementation of the act of desorption. In this case, the presence of random inhomogeneities in the adsorbent material should be considered as the only reason for the appearance of a formally non-integer (see above) desorption order.

Natural candidates for the role of adsorbates participating in the adsorption-desorption processes in an unchanged form are atoms of noble gases, which are practically not able to form molecules. As an example, we mention the article [14] on the kinetics of desorption of argon (and methane) from a substrate of crystalline MgO. It is important that in this case, too, it was possible to isolate the regions of “high” (*θ* > 0.2) and “low” (*θ* < 0.2) fractions of coverage of the MgO surface with argon atoms (compare the discussion of the existence of similar *θ* regions in experiments [6,7,8,9,10,11,12] on H_2_ desorption). Since, surely, no signs of a second-order reaction (corresponding to a hypothetical step of combining argon atoms into molecules) could be detected, it was necessary to admit [14] that the observed kinetics of argon desorption from the MgO surface can be explained, in fact, by the presence of surface defects (steps and kinks [14] or O and Mg vacancies [15]). In this situation, as the initial expression for the rate of argon desorption from the MgO surface in [14,15], naturally, a case of *m* = 1 in Equation (1) was chosen.

However, the solution of Equation (1) in the form of an usual Debye exponential (Equation (2)),
(2)$$\theta (t)=\theta (0)e^{-k_{1}t}$$
with a constant value of $k_{1}$ is insufficient to describe experiments on the desorption kinetics of a number of adsorbates (including argon) in a wide time interval. Therefore, it was proposed in [14,15] to consider the desorption “constant” as a certain quantity depending on *θ* through the activation energy of desorption: $k_{1}=k_{1}^{\left(0\right)}\exp[-E(\theta )/T]$, where *T* is the temperature, $k_{1}^{\left(0\right)}$ is a pre-exponential factor (hereinafter it is convenient to set the Boltzmann constant equal to unity [16]; if necessary, it can always be restored). In this way, an attempt was made to indirectly take into account the role of collective effects in desorption processes, although the whole result [14,15] was reduced, in fact, only to the restoration of function E(θ) from experiments, that used the so-called temperature-programmed desorption techniques. Moreover, it was not possible to establish in [14,15] a deeper connection between the non-Debye kinetics of desorption and the presence of structural defects on the adsorbent surface. It is interesting that the results [14,15,17], in fact, implicitly confirm the physical picture of particle desorption from a structurally disordered surface proposed in the present paper. Indeed, at a high initial fraction of filling of surface states with adsorbate particles, shallow wells, i.e., states with the lowest possible values of the random activation energy of desorption, will be emptied first. As the shallow wells are emptied, i.e., with a decrease in the fraction of filling of surface states, the desorption of particles from ever deeper wells will come into play, which will look like an increase in the effective activation energy of desorption. Qualitatively, a similar picture corresponds to the results of the processing of experiments on temperature-programmed desorption of particles in papers [14,15,17]).

Taking into account the results of experiments [14,15] on the desorption of argon and compounds like CO, methane, etc., it is natural to expect that surface defects will play an important role in the “non-Debye” effects of hydrogen desorption from crystalline adsorbents [6,7,8,9,10,11,12]. Moreover, quantitative arguments can be made in favor of the fact that the inhomogeneities of the surface structure of the adsorbent should be considered as the main reason for the manifestation of the “fractional” kinetics of hydrogen desorption over the entire time range in the experiments [9,11]. So, at each moment of time, we will consider fulfilled the necessary conditions for the recombination of hydrogen atoms into H_2_ molecules, whose desorption will occur according to the equation of the first order “chemical” reaction. In other words, we will use Equation (1) for *m* = 1 as the basis for the description of kinetic curves [9,11] (of course, taking into account the fact that the “constant” of molecular desorption from a structurally inhomogeneous adsorbent will be a random function of 2D coordinates, see below). Wherein the possible manifestation of the case *m* = 2 (when the kinetics of H_2_ desorption is limited by the rate of diffusion of hydrogen atoms with their subsequent recombination), as we will see, will be moved to a region of time beyond the limits of measurements [9,11].

To estimate the time from which the kinetics of hydrogen desorption could, in principle, be described by Equation (1) with *m* = 2, we proceed as follows. First of all, we use the well-known expression for the mean square of the 2D vector of random displacements of an atom over the surface of the adsorbent in the limit of long times (Equation (3)):
(3)$$\langle r^{2}\rangle =4Dt$$
where *D* is the 2D diffusion coefficient (in this case, of hydrogen atoms), and the angle brackets mean the symbol of statistical averaging. The idea of further actions is to compare values from Equation (3) with the introduced above average area *a*^2^ per one hydrogen atom at time *t*. It is more convenient, however, to compare not the quantities themselves, but their time derivatives. Then, if the 4*D* value turns out to be greater than $d(a^{2})/dt$ at the moment *t*, the hydrogen atom, during a random walk on the surface, will be able to visit per unit time the area exceeding the growth rate of the area per one adsorbed atom. In essence, this will mean that, under the formulated condition, the diffusion of hydrogen atoms can be regarded as a “fast” process preceding the association of atoms into H_2_ molecules with their subsequent desorption. As a result, the kinetics of H_2_ desorption will be described by Equation (1) with *m* = 1 up to a certain time *t*_1–2_, after which the stage of desorption kinetics with *m* = 2 can begin. At this stage, the opposite inequality 4*D* < $d(a^{2})/dt$ will already hold; in that case, just the “slow” processes of surface diffusion of hydrogen atoms will be the rate-limiting step determining the probability of second-order reaction and, ultimately, the rate of desorption of the resulting H_2_ molecules (note that under real conditions, the regime at which just the surface diffusion of hydrogen atoms will be the rate-limiting step of molecular desorption may not be achieved at all. Indeed, with typical initial fillings *θ*(0) ~ 10^−1^, H_2_ molecules will form from the nearest atoms, starting from the smallest times, and such recombination will go in parallel with the desorption process. As a result, the desorption of H_2_ molecules, in principle, can be described by the first-order reaction at all times. However, the case where the value of the argument of the logarithm in Equation (4) becomes ~ 1 deserves special attention. Such a possibility occurs at *θ*(0) ~ 10^−4^, if we take for the remaining parameters the same values that were taken above when estimating *t*_1–2_ from Equation (4). Here, however, we will not dwell on the case of such small initial coverings *θ*(0)).

In this way, we obtain equality $4D={\left[d(a^{2})/dt\right]}_{t-t_{1-2}}$, which allows us to estimate the time *t*_1–2_. Given in this equality the relationship $a^{2}(t)=a_{0}^{2}/\theta (t)$ (see above), as well as the solution Equation (2) of Equation (1) at *m* = 1, we arrive at the expression:
(4)$$t_{1–2}=\frac{1}{k_{1}}\ln\left[\frac{4D\theta (0)}{k_{1}a_{0}^{2}}\right].$$

To find the numerical estimate from Equation (4), we use the data on the kinetics of hydrogen desorption from the Si(111)7×7 surface [9] together with the results of measurements of hydrogen surface diffusion on Si(111)7×7 [18]. Assuming, according to [9], *T* = 720 K, *θ*(0) = 0.14, 1/*k*_1_ = 800 s (“half-life”) and taking $D=D_{0}\exp(-E_{a}/T)$, where *D*_0_ = 10^−3^ cm^2^/s and *E_a_* = 1.5 eV [18], we find from Equation (4) the desired estimate *t*_1–2_ ≈ 7500 s. Substituting this value into Equation (2) at the shown parameter values, we obtain for the fraction of coverage at which the diffusion of hydrogen atoms can become the rate-limiting step of desorption, the following estimate: *θ*(7500 s) ~ 10^−5^ (by 3–4 order less than the usually accepted value, delimiting the areas of “high” and “low” *θ* [9]).

We pay attention to the fact that the found time *t*_1–2_ turns out to be approximately one and a half times longer than the maximum (≈5000 s at *T* = 720 K) time interval for measuring the kinetics of hydrogen desorption from the Si(111)7×7 surface [9]. The estimate of $t_{1–2}$ itself, in fact, means that up to *t* ≈ 7500 s the kinetics of hydrogen desorption from the surface of an ideal crystal should be described by the “chemical” reaction of the first-order, and only at *t* > 7500 s would a manifestation of the second-order kinetics be expected [see Equation (1)]. Meanwhile, according to experiments [9], the transition to the “fractional” (*m* ≈ 1.56) order of hydrogen desorption occurs already at *t* ≈ 800 s, which, in essence, is interpreted in [10,11] as cross-over to the second-order desorption. Such a mismatch (almost an order of magnitude in time) of the experimental data [9,11] on hydrogen desorption from the crystalline silicon surface and their interpretation using the model [10,11] (see also [12]) indicates the need for a radical revision of the foundations of this model.

In this paper, the problem of the non-Debye desorption kinetics of particles of various nature—from noble gas atoms (held on the substrate by physical adsorption) to hydrogen (chemically bonding with the adsorbent material)—is considered from the general positions of the theory of thermodynamic fluctuations of physical quantities [16]. In this case, the approach will be based on the first-order kinetic Equation (1) with its “unperturbed” solution in the form of a conventional Debye exponential (Equation(2)), where, however, the rate “constant” will be considered as a random (Gauss-like) function of 2D coordinates at the adsorbent surface. Averaging the exponential of Equation (2) over fluctuations of the rate “constant” by the method proposed for the first time in papers [19,20] will lead to a formula for the non-Debye desorption kinetics (Section 2), which is a theoretical alternative to the model expression with a fractional exponent of desorption [9]. Further (Section 3), using a number of examples (including molecular desorption from both crystalline and amorphous adsorbents), we will demonstrate the possibility of a quantitative interpretation of experiments within the framework of the proposed theory. In the final section, conclusions from the obtained results will be formulated and ways of further development of the proposed theory will be outlined, an additional verification of which could be carried out by setting up purposeful experiments on the adsorption-desorption of molecules on substrates with an inhomogeneous structure.

## 2. The Model of Molecular Desorption from an Inhomogeneous Surface

Before proceeding to the obtained expressions for the rate of non-Debye desorption of molecules from a substrate characterized by some degree of structural disorder, we formulate the main points of the approach under development. Firstly, the disorder is supposed to be frozen (static) and Gaussian. Secondly, if, as a result of the act of desorption, the adsorbate particles leave the substrate in the form of molecules (as in the case of H_2_, see above), the process of recombination of atoms at nearby sites will be considered “fast” in comparison with the subsequent act of molecular desorption. Thirdly, the stage of atomic diffusion over the substrate surface due to which the atoms can meet each other and form a molecule will also be assumed to be “fast” (we will not dwell on the case of extremely small adsorbate coverage of the substrate when just the atomic diffusion step can limit the entire molecular desorption process). For the case of desorption of noble gas atoms, only the first point is essential (cf. the discussion above).

We draw attention to the fact that in a similar formulation we calculated [21] a non-Arrhenius additive to the activation energy of the so-called Henry constant $K_{H}^{\left(\mathrm{fr}\right)}(T)$ characterizing the equilibrium molecular adsorption on a substrate with frozen (the superscript (fr)) disorder. In this case, the resulting expression for $K_{H}^{\left(\mathrm{fr}\right)}(T)$ was obtained in the form [21]:
(5)$$K_{H}^{\left(\mathrm{fr}\right)}(T)=\sqrt{\frac{2\pi \hbar^{2}}{mT^{3}}}\exp\left(\frac{U_{0}}{T}+\frac{\Delta^{2}}{T^{2}}\right).$$
where *m* is the molecular mass, *U*_0_ > 0 is some initial (for an “ideal”, without fluctuations, surface) activation energy, the energy necessary to remove the adsorbed molecule from the potential well on the adsorbent surface back to the bulk adsorbate gas, which is considered ideal. The term $\Delta^{2}/T^{2}$ in brackets of Equation (5) is the desired non-Arrhenius additive to *U*_0_/*T* so that the quantity $\Delta^{2}/T$ means the fluctuation additive to the initial activation energy *U*_0_. Moreover, it is precisely the quantity Δ that will bear averaged information about the distribution function of frozen fluctuations of the activation energy (regarding the relationship between the quantity Δ and the parameters that determine the fluctuation characteristics of a disordered adsorbent, see [21]).

There is a deep connection between the manifestation of fluctuations in the thermodynamics of molecular adsorption-desorption and the specific features of the desorption kinetics from the inhomogeneous substrate. To establish this connection, we write the well-known equation for the rate of change of the relative fraction *θ* of the substrate coverage by the adsorbate molecules (*θ* < 1, see above) at given temperature and pressure *p* (see, e.g., [1]):
(6)$$\frac{d\theta }{dt}=-\frac{\theta }{\tau_{d}}+k_{a}a_{0}^{2}p$$
where $\tau_{d}$ is a characteristic time of desorption, *k_a_* is a rate constant of adsorption (the definition of the area $a_{0}^{2}$ was introduced above). Equation (6) can be obtained from Equation (1) at *m* = 1 by adding the adsorption term and taking the identity $\tau_{d}\equiv 1/k_{1}$ into account (here we are distracting from the possible presence of random inhomogeneities of the adsorbent, assuming $\tau_{d}$ and *k_a_* independent of the coordinates on the substrate surface). Since desorption is an activated process, the characteristic desorption time and the activation energy *U*_0_ appearing in Equation (5) are related by the well-known Arrhenius formula:
(7)$$\tau_{d}=\tau_{d\infty }e^{U_{0}/T}$$
where the pre-exponential $\tau_{d\infty }$ is some molecular time.

Upon reaching a balance between the processes of adsorption and desorption, from Equations (6) and (7) we find the equilibrium (subscript “eq”) numerical density of adsorbate molecules on the adsorbent surface for given *T* and *p*:
(8)$$\theta_{\mathrm{eq}}(T,p)/a_{0}^{2}=k_{a}\tau_{d\infty }e^{U_{0}/T}p\equiv K_{H}(T)p$$
where we used the known definition of the Henry constant (see, e.g., [1,16]). Comparing Equations (8) and (5) (without the non-Arrhenius additive), we arrive at a useful relation,
(9)$$\tau_{d\infty }k_{a}=\sqrt{\frac{2\pi \hbar^{2}}{mT^{3}}}.$$

In fact, relation Equation (9) expresses the rate constant of adsorption in terms of molecular parameters (*m* and $\tau_{d\infty }$) and temperature, taking into account the fact that the adsorption process (in contrast to the desorption process) does not require overcoming the activation barrier.

We now turn to the formulation of an approach that allows us to describe the kinetics of desorption of molecules from a surface, the desorption characteristics of which are random functions of 2D coordinates. We take equality in Equation (2) as the basis of the approach and introduce the local value of the characteristic desorption time:
(10)$$\tau_{d}\{\delta U\}=\tau_{d\infty }\exp\left(\frac{U_{0}+\delta U}{T}\right)$$
where the term δU is a random additive to the activation energy of desorption. In this case, the probabilities of the realization of different values of δU will ultimately be determined by the conditions for the preparation of a structurally inhomogeneous adsorbent. To analyze the thermodynamic equilibrium between the processes of molecular adsorption and desorption on a structurally inhomogeneous substrate, the constant time $\tau_{d}$ should be replaced in Equation (8) by a local (fluctuating) value $\tau_{d}\{\delta U\}$ according to Equation (10). Then, using relation Equation (9) and performing statistical averaging with the distribution function of frozen fluctuations (see [21]), we return to Equation (5). So, generalizing Equation (2) to the case of an adsorbent with a frozen disorder, we will represent the time decay of the averaged (indicated by angle brackets with subscript δU) coverage fraction of such an adsorbent with adsorbate molecules in the form:(11)$$\theta^{\left(\mathrm{fr}\right)}(t)=\theta (0){\langle \exp\left(-t/\tau_{d}\{\delta U\}\right)\rangle }_{\delta U}.$$

The problem of calculating the function $\theta^{\left(\mathrm{fr}\right)}(t)$, in fact, turns out to be similar to that of finding the relaxation function of ionic conductivity for glass-like material [19,20]. Therefore, the averaging of the Debye exponential in Equation (11) for an adsorbent with a Gaussian disorder can be carried out in explicit form by the method developed in [19,20]. Without dwelling on the method of calculations, we present the result (cf. Equation (36) in [20]) in terms of the defined above quantities:(12)$$f(t)\equiv \frac{\theta^{\left(\mathrm{fr}\right)}(t)}{\theta (0)}=\frac{1}{\sqrt{\pi }}\int_{-\infty }^{\infty }du\;\exp\left[-u^{2}-\frac{t}{\tau_{d}}\exp\left(\frac{2\Delta }{T}u\right)\right].$$. It is remarkable that Equation (12) contains only two physically transparent parameters ($\tau_{d}$, Δ), and this is sufficient for the precise approximation of a wide set of experimental data on the non-Debye kinetics of molecular desorption from disordered adsorbents.

At “long” times, Equation (12) really (see below) leads to the non-Debye desorption kinetics (in the absence of disorder, i.e., at Δ = 0, Equation (12) reduces, of course, to usual Debye exponential) (Function (12) has rich analytic properties. A detailed study of its features was carried out in papers [19,20], to which we refer a reader who is interested in the mathematical side of the issue). Naturally, one and the same quantity Δ, which has a fluctuation origin, defines both the non-Arrhenius form of the Henry constant in Equation (5) for the equilibrium adsorption-desorption processes and the non-Debye desorption kinetics in Equation (12) from the disordered adsorbent. In the next Section, Equation (12) will be applied to the quantitative description of the kinetic curves of molecular desorption from the surface of an adsorbent containing frozen structural inhomogeneities.

## 3. Comparison of the Theory with Experimental Data

### 3.1. Non-Debye Kinetics of Hydrogen Desorption from Crystalline Adsorbents Containing Frozen Defects

Apparently, the most convincing measurements of the kinetics of hydrogen desorption from crystalline surfaces were carried out on silicon adsorbents (see [9,11]) containing, in addition to intrinsic structural defects [13], also a certain fraction of impurity atoms (see Introduction). We will consider such inhomogeneities in the adsorbent surface as a source of frozen fluctuations in the local activation energy of desorption of hydrogen molecules. Assuming that the conditions formulated at the beginning of the previous section are satisfied, we will use Equation (12) with allowance for Equation (7) to describe the desorption kinetics of molecules from an inhomogeneous surface, considering the quantities *U*_0_, $\tau_{d\infty }$, and Δ as exhaustive characteristics of the process at a fixed temperature. In this case, naturally, the quantity Δ that carries integrated information about the fluctuation properties of the frozen disorder will play a special role in the manifestation of the effects of the non-Debye kinetics of desorption.

Let us first turn to the experimental [9] temporal dependence of the fraction of hydrogen coverage of a slice of the Si (111)7×7 crystalline surface at *T* = 720 K (points in Figure 1). The solid curve in Figure 1 shows the results of processing this dependence according to Equation (12) for the following parameter values: $\tau_{d}$ = 716 s and Δ = 493 K (taking into account the fact that *θ* (0) = 0.14 [9]). The close agreement between the theoretical curve and experimental data [9] is in favor of our approach describing the desorption kinetics of hydrogen molecules from a structurally disordered surface. It is important that in the entire measurement time interval [9], desorption kinetics can be described by a first-order equation with the Gaussian distribution of local values of desorption activation energies. The fact that up to 5000 s there are no signs of “adding” the second-order processes (*m* = 2 in (1)) means that, if such processes can take place, then, at least, at *t* > 5000 s (the numerical estimate of *t*_1–2_ after Equation (4) is consistent with this conclusion).

In Figure 2, by solid (red) curves we show the results of theoretical (using Equation (12) with the parameters given in Table 1) processing of experimental data [9] (dots) on the kinetics of hydrogen desorption from the same adsorbent Si(111)7×7 at different temperatures and the initial condition *θ*(0) = 0.08. In the inset of Figure 2, we represented an Arrhenius plot of $\tau_{d}$ by Equation (7), whence the values $\tau_{d\infty }$= 1.01 × 10^−14^ s and *U*_0_ = 28,420 K follow (the found values $\tau_{d\infty }$ and *U*_0_ ≈ 2.45 eV are in satisfactory agreement with those given in [9]). Note that, ideally, when processing the experimental dependences in Figure 1 and Figure 2 by our theory, we would have to obtain coincident values of the parameter Δ as the fluctuation characteristic of the frozen disorder for all curves. Meanwhile, in reality, imposing the requirement of the best agreement between theory and experiments [9] at each temperature, we will inevitably get some variation in the values of Δ, which may be due, in particular, to the error of experimental measurements. So, taking $\tau_{d\infty }$= 1.01 × 10^−14^ s and varying *U*_0_ within small limits (*U*_0_ = (28,420 ± 110) K), one can still improve the agreement of the theory with experimental data for the found values of Δ (Figure 2 and Table 1). It is important to emphasize, that a spread for Δ is relatively small (see Table 1).

We turn now to Ref. [11], in which the data of a series of experiments on the hydrogen desorption kinetics from another crystalline adsorbent Si(100)2×1 (points in Figure 3) were reported. The results of the theoretical processing of experimental data [11] at different temperatures and the initial condition *θ*(0) = 0.15 are shown in Figure 3 by solid (red) curves; the corresponding parameter values are given in Table 2. Representing the data for $\tau_{d}$ on an Arrhenius plot (not shown; cf. the inset in Figure 2), we obtain the values $\tau_{d\infty }$= 1.61 × 10^−15^ s and *U*_0_ = (28,200 ± 40) K (which agree satisfactorily with those given in [11]). Moreover, as in the case of experiments [9] (cf. Table 1), there is some variation in the values of the “fluctuation” parameter Δ for different curves [11]. We draw attention to the fact that the mean value of Δ, reconstructed with the help of our theory from experiments [9], is approximately twice as large as the mean value of Δ in experiments [11]. This means that the effects of non-Debye desorption of hydrogen from the Si(111)7×7 surface should be more pronounced than from the Si(100)2×1 surface (this can be easily seen at a qualitative level from a comparison of the shapes of the curves in Figure 2 and Figure 3).

The presence of the line with *c* = 0.99 in Table 2 means that for better agreement between the theory and experiments [11] we modified Equation (12) by introducing the concentration (1–*c*) of “unremovable” adsorbate molecules on the adsorbent:(13)$$f(t)=\frac{c}{\sqrt{\pi }}\int_{-\infty }^{\infty }du\;\exp\left[-u^{2}-\frac{t}{\tau_{d}}\exp\left(\frac{2\Delta }{T}u\right)\right]+(1-c)$$

Physically, such a modification could mean the presence of the adsorbent of a small concentration of deep traps that do not empty at the temperatures of experiments [11]. It has already been noted above that structural inhomogeneities of various nature can be present on the surface of a crystalline adsorbent: intrinsic defects [13] and impurities [8,9]. In this case, it is possible that intrinsic defects (vacancies) are a source of “frozen” fluctuation fields that determine the value of Δ, while impurities play the role of the above-mentioned deep traps (such a question, however, requires special confirmation).

Note that, in principle, numerical estimates of the fundamental quantities $\tau_{d\infty }$ and *U*_0_ included in the parameter $\tau_{d}$ (see Equation (7)), as well as the fluctuation characteristic Δ, could be found using the methods of quantum chemistry and computer simulation (see, for example, review article [22]). This is, however, a separate problem.

Thus, the proposed theory allows us to give a consistent quantitative description of the series of experimental data on the non-Debye kinetics of hydrogen desorption from various crystalline adsorbents with frozen disorder and at different temperatures.

### 3.2. Non-Debye Kinetics of Molecular Desorption from Microporous Adsorbents

The theory represented above gave a possibility to connect the phenomenon of the non-Debye desorption of molecules from a “damaged” crystalline surface with its fluctuation characteristics. On the other hand, there are a number of materials (including zeolites [2,5]), microporous metal-organic materials (see, e.g., [3,4]), which have a complex microstructure with a developed inner surface and are capable of adsorbing various molecules (water, etc.). It is natural to expect that the effects of the non-Debye kinetics of molecular desorption from such compounds should be even more pronounced than in the case of desorption from a crystalline substrate containing frozen defects. Therefore, it makes sense to try to extend the model of non-Debye desorption constructed in the previous Sections to the cases when materials like zeolites act as adsorbents.

As an example, we processed a series of experimental data on the desorption kinetics of *n*-pentane from NaX zeolite granules into a vacuum at temperatures from 273 to 378 K (see Figure 7.3a in the book [2]). In Figure 4, dots show the results of experiments [2], while solid (red) curves represent their theoretical approximation by the Equation (13) with the values of the parameters given in Table 3. Note that the experimental data [2] at each temperature can be quantitatively interpreted in the framework of our theory. However, an Arrhenius plot for the $\tau_{d}$ data can be built by Equation (7) with very big error and crudely gives $\tau_{d\infty }$ ≈ 7.6 s and *U*_0_ = (1530 ± 150) K. As for the value of Δ, its scatter over the entire temperature range turns out to be too significant (Table 3) in order to consider it as a certain fixed fluctuation characteristic of the structure of NaX zeolite. Let us also pay attention to the fact that with a change in temperature, the fraction (1–*c*) of “unremovable” adsorbate molecules varies over a wide range (Table 3).

These circumstances, in essence, mean that fluctuations of the NaX structure can hardly be considered frozen in experiments [2] on the adsorption-desorption of *n*-pentane molecules. The latter ones belong to the type of *n*-alkanes and are chain-like formations with pronounced anisotropy (see, e.g., [23]). As a result, due to the presence of rotational and bending degrees of freedom, the adsorbate molecules when interacting with the walls of the micropores can affect the internal structure of the adsorbent. Such an effect may depend on temperature, for example, in connection with the phenomenon of thermal expansion of the material. Therefore, when describing the non-Debye kinetics of desorption of complex molecules from microporous materials (see [2]), the fluctuation characteristic Δ in Equation (13) should be considered as some effective (possibly temperature-dependent, as in Figure 4) value. Under such conditions, the parameter $\tau_{d\infty }$, of course, loses the meaning of a certain molecular time, but rather determines a characteristic time of the structural relaxation of the porous medium.

In connection with the above, it is useful to mention the review article [24] devoted to methods of molecular simulation of kinetic processes in zeolites. In principle, using such methods, it would be possible to restore the numerical values of the quantities appearing in Table 3 (which, however, is beyond the scope of the present paper).

## 4. Summary and Conclusions

An analysis of the experimental data on the non-Debye kinetics of molecular desorption from structurally disordered adsorbents allows us to conclude that the nature of this kinetics is due, first of all, to random additives to the local activation energy necessary to return the adsorbed molecule back to vacuum. In such a formulation, the question of hydrogen desorption from “damaged” silicon crystals (Si (111)7×7 [9] and Si(100)2×1 [11]) is considered anew using the local Polanyi–Wigner kinetic equation of the first order. Averaging the solution of this equation over the fluctuations (Gaussian) of the local desorption rate constant (based on the approach developed in [19,20]) makes it possible to calculate the resulting time dependence of the coverage of the disordered adsorbent by hydrogen molecules. This dependence (Equation (12)) contains only two fundamental parameters ($\tau_{d}$ andΔ) having transparent physical meaning. At “long” times, the dependence (Equation (12)) decays slower than the conventional Debye exponential, which allows us to quantitatively describe the details of experiments [9,11] on the non-Debye kinetics of hydrogen desorption from adsorbents with frozen disorder.

It should be noted that earlier [11,12] the following assumption was used for modeling the non-Debye kinetics of hydrogen desorption from silicon surfaces. Namely, it was assumed that a first-order process (Debye desorption of H_2_ molecules) should take place at relatively “high” fractions of surface coverage with an adsorbate (in the form of hydrogen atoms). For “low” fractions of coverage, it was assumed that the rate of desorption of H_2_ molecules would be limited by second-order processes, i.e., the rate of formation of molecules from diffusing hydrogen atoms. As the moment of the transition to second-order desorption in [11,12], the time was chosen when the transition from the Debye exponential to the regime with a fractional desorption exponent [9] was observed on the kinetic curves.

However, above, based on the results of experiments [18], we presented numerical arguments in favor of the fact that the time from which the rate of desorption of H_2_ molecules from the Si(111)7×7 surface will be determined just by the diffusion of hydrogen atoms, is approximately an order of magnitude longer than the time of transition to the regime with a “fractional” desorption exponent [9]. This conclusion immediately compels us to consider the non-Debye desorption of hydrogen from silicon adsorbents [9,11] as a manifestation, first of all, of the structural disorder of such adsorbents (about the presence of a significant density of defects such as vacancies on the Si(001) surface, see [13]). From here, it follows that for adsorbent-adsorbate systems in which the stage of the formation of molecules from adsorbate atoms could be eliminated from the very beginning, the whole effect of the non-Debye desorption would inevitably have to be attributed to the structural disorder of the adsorbent.

This state of affairs makes us draw attention to the need for special experiments in order to obtain detailed information about the non-Debye kinetics of the desorption of precisely atoms from a substrate containing frozen defects. As natural adsorbates for such experiments, it is necessary to consider, first of all, atoms of noble gases. To date, the effects of non-Debye desorption of argon [14] (as well as methane [14] and CO [15]) have been recorded from a MgO(100) crystalline substrate containing surface defects (see also [17] concerning the desorption of a number of substances, including atoms of noble gases, from such adsorbents as graphene and amorphous solid water). On the whole, however, the question of the non-Debye desorption of substances bound to the substrate by the forces of physical adsorption is relatively little studied experimentally. For example, the lack of information about the desorption curves of noble gas atoms from disordered adsorbents in real-time makes it difficult to obtain information on the structural and kinetic parameters of the adsorbent-adsorbate system. Meanwhile, just experiments on non-Debye desorption of argon-type atoms could serve as a source of the most reliable information on the parameters of the kinetic Equation (12), since second-order desorption effects, in this case, would be excluded by definition.

Regarding the problem of non-Debye desorption of hydrogen from “damaged” crystalline adsorbents, experiments in the field of ultra-small surface coverage with an adsorbate would be of interest here, when the process of combining of atoms into H_2_ molecules could become the stage limiting the desorption kinetics (see the discussion of this question above).

Finally, another important consequence of the proposed theory is the establishment of a fundamental relationship between the appearance of the “non-Arrhenius” contribution in Henry’s law (as an equilibrium characteristic of adsorption-desorption processes) and the deviation of the kinetics of molecular desorption from the Debye one for structurally disordered adsorbents. In this case, the fact of a joint description of the thermodynamic (the “non-Arrhenius” Henry constant) and kinetic (the non-Debye desorption) experimental data for a specific system when using one and the same series of parameters (including the fluctuation characteristic Δ) could be considered as the evidence of the self-consistency of the proposed theory. It seems that verification of this possibility with the help of specially designed experiments could be of fundamental importance.

## Figures and Tables

**Figure 1 molecules-25-03662-f001:**
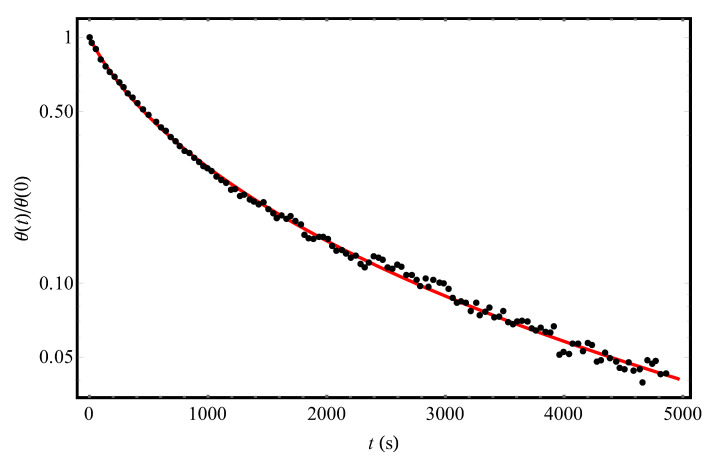
The experimental temporal dependence (points, [9]) of the reduced hydrogen coverage *θ*(*t*)/*θ*(0) of the Si(111)7×7 adsorbent at *T* = 720 K and the initial condition *θ*(0) = 0.14. The solid (red) curve represents the theoretical processing of the data [9] by Equation (12) with parameters Δ = 493 K and $\tau_{d}$ = 716 s.

**Figure 2 molecules-25-03662-f002:**
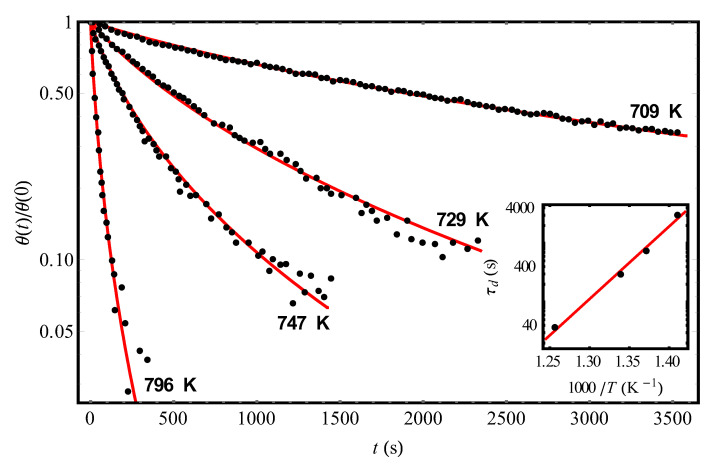
The experimental temporal dependences (points, [9]) of the reduced hydrogen coverages *θ*(*t*)/*θ*(0) of the Si(111)7×7 adsorbent at different temperatures and the initial condition *θ*(0) = 0.08. The solid (red) curves represent the theoretical processing of the data [9] by Equation (12) with parameters Δ and $\tau_{d}$ pointed out in Table 1. The inset demonstrates an Arrhenius plot of the found $\tau_{d}$ values by Equation (7) with $\tau_{d\infty }$ = 1.01 × 10^−14^ s and *U*_0_ = 28,400 K.

**Figure 3 molecules-25-03662-f003:**
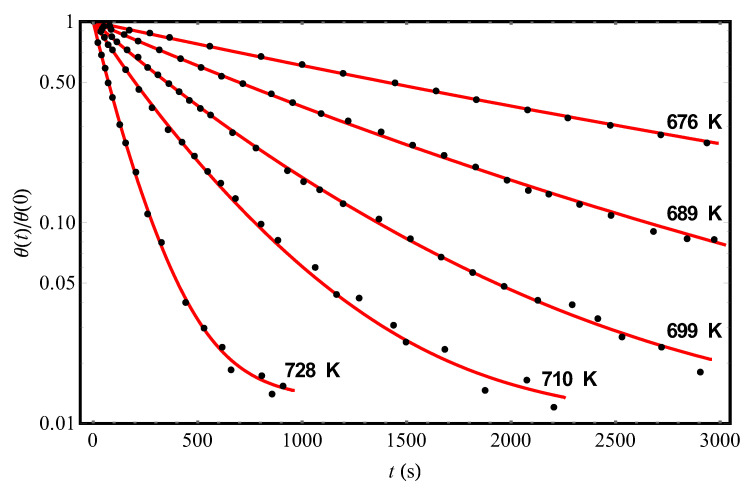
The experimental temporal dependences (points, [11]) of the reduced hydrogen coverages *θ*(*t*)/*θ*(0) of the Si(100)2×1 adsorbent at different temperatures and the initial condition *θ*(0) = 0.15. The solid (red) curves represent the theoretical processing of the data [11] by Equation (13) with parameters Δ, *c*, and $\tau_{d}$ pointed out in Table 2.

**Figure 4 molecules-25-03662-f004:**
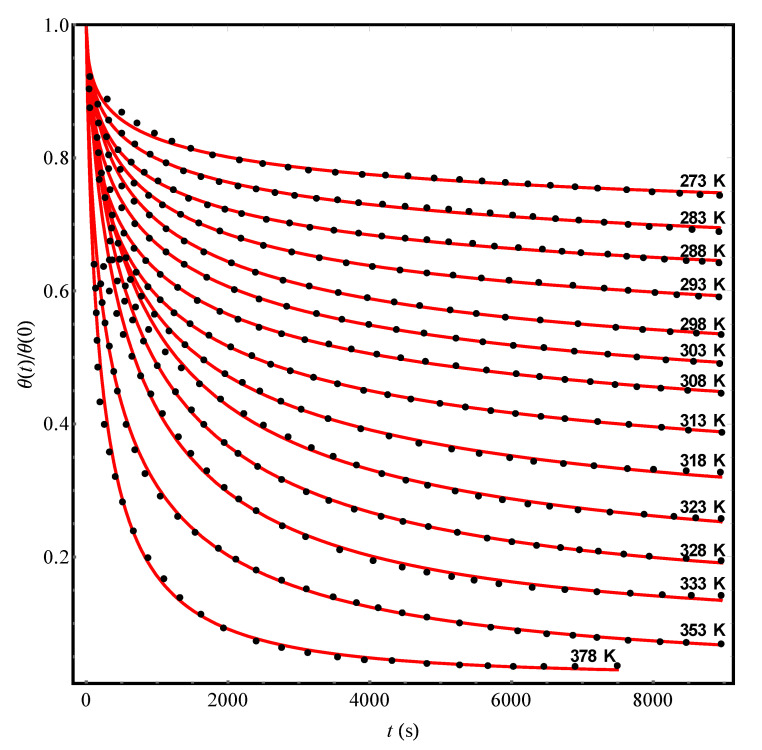
The experimental reduced kinetic curves (points, [2]) of the *n*-pentane desorption from pellets of NaX zeolite at different temperatures. The solid (red) curves represent the theoretical processing of the data [2] by Equation (13) with parameters Δ, *c*, and $\tau_{d}$ pointed out in Table 3.

**Table 1 molecules-25-03662-t001:** The results of the theoretical processing of experimental data [9] on the kinetics of hydrogen desorption from the Si(111)7×7 surface at different temperatures and the initial condition *θ*(0) = 0.08 (Figure 2). The values of the characteristic parameters $\tau_{d}$ and Δ were found by approximating the data [9] according to Equation (12) with the use of the least squares method.

***T* (K)**	709	729	747	796
**Δ (K)**	470	420	460	460
***τ_d_* (s)**	2980	741	299	37.2

**Table 2 molecules-25-03662-t002:** The results of theoretical processing of experimental data [11] on the kinetics of hydrogen desorption from the Si(100)2×1 surface at different temperatures and the initial condition *θ*(0) = 0.15 (Figure 3). The values of the characteristic parameters $\tau_{d}$, Δ, and *c* were found by approximating the data [11] according to Equation (13) with the use of the least squares method.

***T* (K)**	676	689	699	710	728
**Δ (K)**	180	180	197	200	200
***τ_d_* (s)**	2030	1010	510	290	105
***c***	0.99	0.99	0.99	0.99	0.99

**Table 3 molecules-25-03662-t003:** The results of the theoretical processing of experimental data [2] on the kinetics of *n*-pentane desorption from pellets of NaX zeolite at different temperatures (Figure 4). The values of the characteristic parameters $\tau_{d}$, Δ, and *c* were found by approximating the data [2] according to Equation (13) with the use of the least squares method.

***T* (K)**	273	283	288	293	298	303	308
**Δ (K)**	580	575	530	470	440	490	525
***τ_d_* (s)**	1489	1668	1313	1328	1360	1299	1127
***c***	0.33	0.40	0.44	0.49	0.55	0.61	0.66
***T* (K)**	313	318	323	328	333	353	378
**Δ (K)**	480	415	345	330	320	425	345
***τ_d_* (s)**	1061	1160	1226	1087	879	540	293
***c***	0.71	0.77	0.82	0.87	0.91	0.98	0.98

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
