# Peer review of "Long-Time Non-Debye Kinetics of Molecular Desorption from Substrates with Frozen Disorder"

_molecules, 2020, doi:10.3390/molecules25163662_

Round 1
Reviewer 1 Report
The paper presents a general description of the kinetics of hydrogen desorption from substrates characterized by some degree of structural disorder, from crystalline Silicon containing frozen defects to zeolites. The proposed formulation attributes the deviations observed from the first-order Debye exponential desorption and the necessity of introducing fractional orders in the kinetic equation to the presence of surface inhomogeneities rather than to the change in the desorption mechanism at low fractions of coverage. The close expression obtained for the rate allows an accurate description of different experiments.
The paper is well organized and clearly written, giving in the Introduction an overview of the general problem with the relevant references in the literature, and is suitable for publication in Molecules.
The paper is interesting and the fact that the parameters in the working equations have a clear physical meaning is for sure appealing. I would like to suggest the authors to include a brief discussion on this issue. Are the modern methods in quantum chemistry able to give a theoretical estimation of the parameters or to identify loci on the surface that trap the adsorbed species, originating the (1-c) fraction? Is it related to the well depth in the chemisorption interaction potential?
Typos:
paragraph after Eq.(5) ".., the quantity Delta^2/T is the desired" should be ".., the quantity Delta^2/T^2 is the desired"
(latex notation for math)
Reviewer 2 Report
In this paper the authors describe a theoretical approach to hydrogen desorption from solid surfaces. While I think that the paper is well written and the outcome fits nicely to the experimental results, I still have one question. Could the effects not also be explained by changes in the interactions between the adsorbed species as a fuction of coverage. MAybe the authors want to at least comment on this possibility in the manuscript.
Reviewer 3 Report
The authors propose a modified theory of adsorption/desorption that takes into account frozen fluctuations in the activation energy which seem to play a crucial role in the non-Debye kinetics. The work is well explained and the theory is corroborated by experimental data of simple adsorbate processes on crystalline and amorphous solids.
I have several minor comments:
abstract - "spoiled" is never defined in this work
page 1 - "Refs. 3 and 4" do not exist - please give author names and years
page 15 - "genetic relationship" - perhaps the authors have the wrong word here, it seems "genetic" is inappropriate.
Lastly, are the authors aware of the many amorphous porous solids used for adsorption (such as ordered mesoporous silica, zeolite-templated carbon, etc.) that have somewhat regular pore-to-pored structure? I recommend improving the introduction with a better overview of work on MOFs (instead of just 3 select papers) and some perspective on non-crystalline solids. There are also large review papers overviewing the field of adsorption that might be applicable.
